# *Cyclopia intermedia* (Honeybush) Induces Uncoupling Protein 1 and Peroxisome Proliferator-Activated Receptor Alpha Expression in Obese Diabetic Female db/db Mice

**DOI:** 10.3390/ijms24043868

**Published:** 2023-02-15

**Authors:** Babalwa Unice Jack, Pritika Ramharack, Christiaan Malherbe, Kwazi Gabuza, Elizabeth Joubert, Carmen Pheiffer

**Affiliations:** 1Biomedical Research and Innovation Platform, South African Medical Research Council, Cape Town 7505, South Africa; 2Pharmaceutical Sciences, School of Health Sciences, University of KwaZulu-Natal, Westville Campus, Durban 4001, South Africa; 3Plant Bioactives Group, Post-Harvest and Agro-Processing Technologies, Agricultural Research Council (ARC), Infruitec-Nietvoorbij, Stellenbosch 7599, South Africa; 4Department of Food Science, University of Stellenbosch, Stellenbosch 7602, South Africa; 5Centre for Cardio-Metabolic Research in Africa (CARMA), Division of Medical Physiology, Faculty of Medicine and Health Sciences, University of Stellenbosch, Cape Town 7505, South Africa; 6Department of Obstetrics and Gynaecology, Faculty of Health Sciences, University of Pretoria, Pretoria 0001, South Africa

**Keywords:** *Cyclopia intermedia*, db/db mice, brown adipose tissue, uncoupling protein 1, peroxisome proliferator activator receptor alpha, hepatic fat accumulation, molecular docking

## Abstract

Previously, we reported that a crude polyphenol-enriched fraction of *Cyclopia intermedia* (CPEF), a plant consumed as the herbal tea, commonly known as honeybush, reduced lipid content in 3T3-L1 adipocytes and inhibited body weight gain in obese, diabetic female leptin receptor-deficient (db/db) mice. In the current study, the mechanisms underlying decreased body weight gain in db/db mice were further elucidated using western blot analysis and in silico approaches. CPEF induced uncoupling protein 1 (UCP1, 3.4-fold, *p* < 0.05) and peroxisome proliferator-activated receptor alpha (PPARα, 2.6-fold, *p* < 0.05) expression in brown adipose tissue. In the liver, CPEF induced PPARα expression (2.2-fold, *p* < 0.05), which was accompanied by a 31.9% decrease in fat droplets in Hematoxylin and Eosin (H&E)-stained liver sections (*p* < 0.001). Molecular docking analysis revealed that the CPEF compounds, hesperidin and neoponcirin, had the highest binding affinities for UCP1 and PPARα, respectively. This was validated with stabilising intermolecular interactions within the active sites of UCP1 and PPARα when complexed with these compounds. This study suggests that CPEF may exert its anti-obesity effects by promoting thermogenesis and fatty acid oxidation via inducing UCP1 and PPARα expression, and that hesperidin and neoponcirin may be responsible for these effects. Findings from this study could pave the way for designing target-specific anti-obesity therapeutics from *C. intermedia*.

## 1. Introduction

Obesity and its associated metabolic complications such as type 2 diabetes (T2D), non-alcoholic fatty liver disease (NAFLD), and cardiovascular disease (CVD) present a significant global public health concern [1]. In 2016, 13% of adults worldwide were obese [2], with projections of 20% by 2025, unless effective interventions are developed [3]. The burden of obesity is disproportionately higher in women compared to men, with several epidemiological studies reporting higher prevalence in women than in men [3,4,5]. Current approaches for the treatment of obesity include dietary interventions such as low-calorie diets, increased physical activity such as exercise, pharmacological therapy, and surgical procedures [6,7]. Unfortunately, their therapeutic effects against obesity have not been satisfactory, urging the need to explore indigenous medicinal plants for their therapeutic potential in treating modern society diseases such as obesity.

*Cyclopia* is an endemic South African plant genus and several of the species have traditionally been processed and consumed as herbal tea, commonly known as honeybush tea [8]. In addition, anecdotal evidence suggests that drinking infusions of honeybush tea has restorative properties and can stimulate appetite [9], implying its role in improving appetite and thus regulating energy metabolism and obesity. Over the years, this herbal tea has gained popularity on local and global markets partly due to its pleasant aroma and taste, and partly because of mounting scientific evidence of various beneficial health effects [10], including potential anti-obesity properties. Accordingly, recent studies showed that extracts of three *Cyclopia* species exhibit anti-obesity effects [11,12,13]. Aqueous extracts of *C. maculata* and *C. subternata* were shown to inhibit lipid and triglyceride accumulation and decrease peroxisome proliferator-activated receptor gamma 2 (PPARγ2) expression in 3T3-L1 adipocytes [11]. These extracts also induced lipolysis and the expression of hormone sensitive lipase (HSL) and perilipin in 3T3-L1 adipocytes [12]. More recently, our focus shifted to a crude polyphenol-enriched fraction, prepared from *C. intermedia* (CPEF), demonstrating that CPEF decreased the lipid content of 3T3-L1 adipocytes, while also increasing *Hsl* and uncoupling protein 3 (*Ucp3*) gene expression in these cells [13]. Additionally, we showed that treatment of obese, diabetic female leptin receptor-deficient db/db mice with CPEF inhibited body weight gain without affecting food and water intake, nor causing changes in glucose tolerance [13]. Furthermore, CPEF did not alter the expression of genes involved in lipid metabolism, glucose homeostasis, and thermogenesis in subcutaneous and visceral white adipose tissue (WAT) of db/db mice [14], warranting further studies to elucidate the mechanisms underlying its anti-obesity effects.

Increasing thermogenesis in brown adipose tissue and stimulating fatty acid oxidation have emerged as potential therapeutic targets for anti-obesity intervention strategies [15]. A considerable amount of evidence suggests that uncoupling protein 1 (UCP1) is at the core of brown adipose tissue thermogenesis and systemic energy homeostasis [16], while peroxisome proliferator-activated receptor alpha (PPARα) is the key master of lipid metabolism, mainly involved in regulating fatty acid oxidation and its key genes. As such, polyphenols that are able to target UCP1 and PPARα, and increase their expression, thus inducing thermogenesis or stimulating fatty acid oxidation, have attracted considerable interest as novel strategies for the treatment of obesity [17,18,19]. In this study, we hypothesised that the previously reported anti-obesity effects of CPEF in db/db mice [13] are due to the induction of UCP1 and PPARα expression, and accordingly quantified UCP1 and PPARα expression, and protein oxidation levels in brown adipose tissue. In addition, we measured PPARα expression, lipid peroxidation, and protein oxidation in the liver of the CPEF-treated db/db mice. Hepatic fat accumulation was quantified in hematoxylin and eosin-stained liver sections. Furthermore, we used in silico molecular docking and ligand interaction plot analysis to predict polyphenols that might be responsible for the increased UCP1 and PPARα expression in CPEF-treated db/db mice.

## 2. Results

### 2.1. Treatment with CPEF Increases UCP1 and PPARα Expression in Brown Adipose Tissue

The expression of UCP1 in the brown adipose tissue was approximately 3.4-fold higher in obese, diabetic female db/db mice treated with CPEF compared to untreated db/db control mice (99.45 ± 65.45% vs. 29.37 ± 18.82%, *p* < 0.05) (Figure 1A,B). In addition, CPEF treatment increased PPARα expression 2.6-fold compared to untreated db/db control mice (62.74 ± 35.96% vs. 24.25 ± 15.97%, *p* < 0.05) (Figure 1A,C).

### 2.2. Treatment with CPEF Decreases Hepatic Lipid Accumulation

Histological analysis of Hematoxylin and Eosin (H&E)-stained liver sections showed reduced intracellular fat vacuoles in CPEF-treated db/db mice compared to their untreated db/db controls (Figure 2A). Quantification using Image J showed an approximately 31.9% decrease in lipid accumulation in CPEF-treated mice (17.77 ± 5.88% vs. 26.11 ± 6.37%, *p* < 0.001) (Figure 2B).

### 2.3. Treatment with CPEF Increases Hepatic PPARα Expression

A 2.2-fold increase in PPARα expression was observed in the livers of CPEF-treated db/db mice compared to their untreated db/db control mice (158.60 ± 35.77% vs. 72.47 ± 38.70%, *p* < 0.01) (Figure 3A,B).

### 2.4. Effect of CPEF Treatment on Lipid Peroxidation and Protein Oxidation

Lipid peroxidation levels tend to be lower in CPEF-treated db/db mice compared to untreated db/db control mice, given a 1.6-fold lower hepatic MDA content (0.43 ± 0.22 vs. 0.68 ± 0.22 µmol/mg, *p* = 0.07) (Figure 4). Lipid peroxidation could not be quantified in the brown adipose tissue due to the limited amount of tissue sample available. Protein carbonylation in the brown adipose tissue was 1.2-fold lower in CPEF-treated db/db mice compared to untreated db/db control mice (0.29 ± 0.07 vs. 0.36 ± 0.05 nmol/mg, *p* = 0.06) (Figure 5A), whereas no differences were observed in the liver (*p* = 0.59) (Figure 5B).

### 2.5. Prediction of Compounds Responsible for the Increased Expression of UCP1 and PPARα, Elicited through CPEF

Figure 6 provides the binding scores of CPEF phenolic compounds and CL-316,243 (used as a positive control for UCP1 ligand) for the UCP1-complex. When compared to CL-316,243 (−7.6 kcal/mol), the CPEF phenolic compounds, 3-β-d-glucopyranosyl-4-*O*-β-d-glucopyranosyliriflophenone (−8.1 kcal/mol), 3-β-d-glucopyranosyliriflophenone (−8.0 kcal/mol), hesperidin (−8.8 kcal/mol), and neoponcirin (−7.9 kcal/mol) demonstrated improved binding affinities to UCP1, with the hesperidin-UCP1 complex exhibiting the highest binding score. The mangiferin-UCP1 complex (−7.6 kcal/mol) exhibited the same binding score as the CL-316,243-UCP1 complex (−7.6 kcal/mol), while the binding affinities for vicenin-2 (−7.2 kcal/mol) and isomangiferin (−7.0 kcal/mol) were less than that of CL-316,243 (Figure 6).

For the PPARα complex (Figure 6), hesperidin (−8.4 kcal/mol) showed similar binding affinities to fenofibrate (−8.4 kcal/mol, used as a positive control for PPARα ligand), whilst neoponcirin bound to PPARα with a docking score of −9.8 kcal/mol. The binding score of 3-β-d-glucopyranosyl-4-*O*-β-d-glucopyranosyliriflophenone (−6.3 kcal/mol), 3-β-d-glucopyranosyliriflophenone (−6.3 kcal/mol), isomangiferin (−6.0 kcal/mol), mangiferin (−4.5 kcal/mol), and vicenin-2 (−2.9 kcal/mol) were lower than that of the fenofibrate-PPARα complex (Figure 6).

To further validate the docking scores, ligand interaction plots were analysed (Table 1) to elucidate the intermolecular interactions that stabilised the above-mentioned compounds within the active sites of UCP1 and PPARα. When compared to CL-316,243, interaction plot analysis revealed that the optimally docked UCP1 compounds of CPEF formed conserved intermolecular interactions with Lys38, Val39, Arg140, Gly45, Pro179, and Phe240 (Table 1). These conserved residues indicate that the compounds, hesperidin, 3-β-d-glucopyranosyl-4-*O*-β-d-glucopyranosyliriflophenone, 3-β-d-glucopyranosyliriflophenone, and neoponcirin interact with UCP1 in a similar binding mode as CL-316,243. It was interesting to note that hesperidin formed the greatest number of hydrogen bonds with the surrounding residues (average bond length of 2.91 Å), thus indicating increased stability with the active site (Table 1). It was also interesting to note that Glu46 of UCP1 formed three stabilising hydrogen bonds with the hydroxyl groups of hesperidin (Table 1).

The PPARα interaction plot analysis (Table 1) revealed conserved intermolecular interactions in the neoponcirin, hesperidin, and fenofibrate complexes (Thr279, Met320, Leu321, Cys276, Ile354, Phe273, His440, Phe318, Ile317, Met355, and Ser280). The binding affinities of hesperidin and neoponcirin were validated by the hydrophobic interactions generated with 19 surrounding PPARα residues, compared to the 9 interacting residues identified in the fenofibrate-PPARα complex (Table 1).

## 3. Discussion

Exploring the potential of plant polyphenols to serve as anti-obesity therapeutics is attracting global interest. Previously, we reported that a crude polyphenol-enriched fraction of *C. intermedia* (CPEF) exhibited anti-obesity effects in vitro and in vivo [13], however, the mechanisms underlying these effects were not elucidated. In the present study, we demonstrated that treatment of obese, diabetic db/db mice with CPEF increased the expression of UCP1 and PPARα in the brown adipose tissue, which was accompanied by decreased lipid accumulation and increased PPARα expression in the liver. Moderate improvements in oxidative stress were also observed in these tissues.

Compared to the control db/+ mice, decreased expression of UCP1 in the brown adipose tissue of obese, diabetic db/db mice, was restored after CPEF treatment. UCP1 is an inner membrane protein that uncouples the mitochondrial respiratory chain from oxidative phosphorylation by catalysing the leak of protons across the mitochondrial inner membrane, inhibiting adenosine triphosphate production and dissipating energy as heat [20]. Decreased UCP1 expression and thermogenesis are associated with obesity [21,22], thus, the search for compounds, including *C. intermedia* polyphenols, that are able to upregulate UCP1 expression and potentially induce thermogenesis [17,18,23] is escalating. Previously, we provided some evidence to suggest that CPEF may increase *Ucp1* messenger RNA (mRNA) in subcutaneous and visceral WAT of db/db mice, although the increase was small and not statistically significant [14]. Thus, a follow-up study to evaluate the “WAT browning” potential of CPEF and *C. intermedia* polyphenols was warranted. This entailed quantifying the expression of beige-specific molecular markers in WAT, as previously reported for other phenolic compounds [24].

The expression of PPARα was increased in the brown adipose tissue of CPEF-treated db/db mice. In brown adipose tissue, PPARα agonists directly induced *Ucp1* mRNA transcription, possibly via PPARγ Coactivator 1α (*Pgc-1α*) induction and PRD1-BF1-RIZ1 homologous domain-containing 16 (*Prdm16*) gene expression [25,26]. Thus, our results support an important role of PPARα in regulating UCP1 and thermogenesis in brown adipose tissue [27]. CPEF treatment also increased hepatic PPARα expression in db/db mice, accompanied by decreased lipid accumulation. Low expression levels of PPARα are associated with non-alcoholic fatty liver disease (NAFLD) in obese subjects [28,29,30,31]. Hepatocyte-specific knockdown of PPARα leads to hepatic steatosis by impairing hepatic and whole-body fatty acid homeostasis and increasing hepatic and plasma triglyceride, free fatty acid, and cholesterol levels [32]. Fenofibrate, a selective PPARα agonist, stimulates hepatic β-oxidation and reverses hepatic steatosis in high-cholesterol and fructose-enriched diet fed models [33,34].

Others have similarly reported that plant polyphenols may offer potential to ameliorate NAFLD by reducing fat storage in the liver, inhibiting inflammation, activating autophagy, increasing the expression of genes involved in lipid oxidation including PPARα, and modulating mitochondrial bioenergetics and lipogenesis [35,36,37,38,39,40,41,42]. The xanthone, mangiferin, and the flavanone, hesperidin, both present in substantial quantities in CPEF [13], have been shown to improve hepatic steatosis by reducing the accumulation of lipid droplets and increasing the expression of genes and proteins involved in lipid oxidation [39,41,42], further supporting the ameliorative properties of CPEF against obesity and hepatic lipid accumulation. Future studies measuring hepatic steatosis using techniques such as Oil Red O staining or hepatic triglyceride quantification [43] in response to CPEF treatment are warranted.

The antioxidant properties (i.e., decreased intracellular reactive oxygen species production, increased antioxidant enzyme activity, and reduced oxidative DNA damage, protein oxidation, and lipid peroxidation) of *Cyclopia*, mangiferin and hesperidin, are widely reported [44,45,46,47,48,49]. In our study, CPEF moderately improved lipid peroxidation and protein oxidation, the markers of oxidative stress damage. Similarly, other studies reported improvements in lipid peroxidation in the liver of obese rodents treated with hesperidin [50,51], while there are no reports on brown adipose tissue.

As mangiferin and hesperidin were previously shown to potentially increase UCP1 expression and induce thermogenesis, it would be worthwhile to evaluate the efficacy of the other CPEF polyphenols on inducing UCP1 expression and thermogenesis. However, using high performance counter-current chromatography (HPCCC) fractionation to separate CPEF into four major fractions, predominantly containing 3-β-d-glucopyranosyl-4-*O*-β-d-glucopyranosyliriflophenone (fraction 1), hesperidin (fraction 2), mangiferin (fraction 3), and neoponcirin (fraction 4), we previously showed that these fractions exhibited varying anti-obesity effects, but they were less effective than CPEF [52]. As such, the elucidation of the phenolic compounds that might be responsible for the increased expression of UCP1 and PPARα in CPEF-treated db/db mice was assessed, using molecular docking analysis. Hesperidin and neoponcirin had the highest binding affinities for UCP1 and PPARα, respectively. The docking scores of hesperidin and neoponcirin were even higher than CL-316,243 and fenofibrate, which are commonly used as experimental controls to activate UCP1 and PPARα expression, respectively [53,54]. This was validated by stabilising intermolecular interactions within the active sites of UCP1 and PPARα, when complexed with these compounds. Based on the molecular docking and ligand interaction plot analyses it can be deduced that hesperidin and neoponcirin might be responsible for exerting the increased UCP1 and PPARα expression in response to treatment with CPEF. 3-β-d-glucopyranosyl-4-*O*-β-d-glucopyranosyliriflophenone and 3-β-d-glucopyranosyliriflophenone also had higher binding affinities for UCP1 than CL-316,243. Further investigations, such as biological experiments to validate in silico molecular docking analysis data are necessary and, in this case, to deduce that the increased expression levels of UCP1 and PPARα by CPEF treatment are mediated by hesperidin and neoponcirin.

A strength of our study is the use of obese, diabetic female db/db mice. Despite the disproportionally higher rates of obesity in females compared to males [3,4], the majority of obesity studies in rodents are conducted in males due to concerns that the female estrous cycle makes female models intrinsically more variable than male models [55,56]. Gender-specific metabolic profiles in response to treatment have been reported in animal models [56], therefore it is important that the effects of the CPEF extract observed in this current study should be confirmed in male db/b mice. The lack of positive controls such as β3-adrenergic receptor (CL-316,243), PPARα (fenofibrate), and PPARγ (rosiglitazone) agonists with known potential to activate UCP1 and PPARα expression, induce thermogenesis and fatty acid oxidation, and demonstrate anti-obesity effects in rodents is another limitation [57,58,59]. Future studies to assess brown adipose tissue morphology and elucidate the mechanisms that underlie UCP1 induction in BAT are warranted. In addition to UCP1 expression, quantification of other thermogenic markers such as PRDM16, PGC-1α, neuregulin 4 (NRG4), and fibroblast growth factor 21 (FGF21) is warranted.

## 4. Materials and Methods

### 4.1. Preparation of CPEF

CPEF is the organic fraction, obtained by liquid-liquid partitioning between *n*-butanol and water of a freeze-dried, 40% methanol-water extract of unfermented *Cyclopia intermedia*, as described previously [13]. This includes quantification of the major phenolic compounds in CPEF; the benzophenones, 3-β-d-glucopyranosyl-4-*O*-β-d-glucopyranosyliriflophenone, and 3-β-d-glucopyranosyliriflophenone; the xanthones, mangiferin, and isomangiferin; the flavanones, hesperidin, neoponcirin, and eriodictyol-*O*-deoxyhexose-*O*-hexose; and the flavone, vicenin-2. The xanthones and flavanones were present at the highest concentrations (≥ 2 g/100 g CPEF) [13].

### 4.2. Animals and Treatment Protocol

The animal study has been described previously and only the highest CPEF treatment dose (351.5 mg/kg body weight) was used in this study, as it significantly reduced body weight gain compared to the lower dose (70.5 mg/kg body weight) [13]. Briefly, six-to-seven-week-old female homozygous C57BL/KsJ-Lepr^db/db^ mice and their lean heterozygous counterparts, C57BL/KsJ-Lepr^db/+^ mice were purchased from Jackson Laboratory (Bar Harbor, ME, USA) and housed at the Primate Unit and Delft Animal Centre (PUDAC) of the South African Medical Research Council (Tygerberg, South Africa), in individual cages, under controlled environmental conditions (23–25 °C; ± 55% humidity; 15–20 air changes per hour; 12-h light/dark cycle). The animals were provided with a standard laboratory diet and water, ad libitum. A total of 21 mice were divided into three groups (*n* = 6–8) as follows: normal control (db/+) and the obese, diabetic control (db/db) mice, which were treated with a vehicle control (1% DMSO solution prepared in distilled water); as well as the db/db mice group administered with CPEF (351.5 mg/kg body weight prepared in 1% DMSO) by oral gavage (0.2–0.5 mL volume per body weight) for 28 days [13]. On completion of the treatment period, the mice were anesthetised by inhalation of 2% fluothane and 98% oxygen (AstraZeneca Pharmaceuticals, Johannesburg, South Africa). Liver tissue was collected for histological analysis, while interscapular brown adipose tissue (iBAT) [60] and liver tissue were snap frozen in liquid nitrogen for analysis of protein expression and oxidative stress markers. The study was approved by the Ethics Committee for Research on Animals, South African Medical Research Council (ECRA approval number: 10/13) and the Research Ethics Committee: Animal Care and Use of Stellenbosch University (approval number: SU-ACUM13-00028), and experiments were conducted in accordance with the internationally accepted principles for laboratory animal use and care.

### 4.3. Histology

After termination and excision, liver tissue sections were immediately fixed in 10% formalin (Merck-Millipore, Billerica, MA, USA), processed overnight using a Leica TP 1020 automated processor (Leica Biosystems, Buffalo Grove, IL, USA), and thereafter, embedded in paraffin wax blocks. Paraffin-embedded tissues were sliced into 5-μm sections, fixed onto aminopropyltriethoxysilane coated glass slides (Sigma-Aldrich, St. Louis, MO, USA), then deparaffinised, rehydrated, and stained with hematoxylin and eosin (H&E) (Merck-Millipore). Ten stained liver sections were randomly captured using a Nikon Eclipse Ti inverted microscope (Tokyo, Japan) and NIS Elements imaging software at 20× magnification. The lipid droplets in H&E-stained liver sections were quantified using ImageJ software (Version j 1.52r, US National Institutes of Health, Bethesda, MD, USA) [61], which calculates the area occupied by the fat vacuoles relative to the total area of the liver section.

### 4.4. Protein Extraction and Western Blot Analysis

Approximately 100 mg of the snap frozen iBAT and liver tissues were homogenised in tissue lysis buffer (Tissue Extraction Reagent I, Invitrogen, Carlsbad, CA, USA) supplemented with 1 mM phenylmethane sulfonyl fluoride (PMSF), protease (2 tablets per 100 mL lysis extraction reagent) and phosphatase (10 tablets per 100 mL lysis extraction reagent) inhibitor tablets (Roche Diagnostics, Basel, Switzerland), using a Tissue lyser and pre-cooled adapters (Qiagen, Hilden, Germany) at 25 Hz for 1 min. Cells were homogenised by alternating between 1 min in the Tissue lyser and 1 min on ice, repeated 5 times for the iBAT and 3 times for the liver tissues. Homogenates were centrifuged at 15,890 × *g* for 15 min at 4 °C, whereafter the supernatants were collected. This step was repeated twice. Protein concentrations were measured using the RC DC kit (BioRad Laboratories Inc., Hercules, CA, USA), which is compatible with reducing agents and detergents present in the tissue lysis buffer. Thirty micrograms of heat denatured proteins were resolved by 12% sodium dodecyl sulfate (SDS)-polyacrylamide gel electrophoresis and transferred to polyvinylidene fluoride (PVDF) membranes using a Trans-Blot^®^ Turbo™ Transfer System (BioRad). Membranes were blocked, using 5% fat-free milk in Tris-buffered saline (pH 7.2) supplemented with 0.1% Tween^®^ 20 (TBST) applied at room temperature for up to 3 h, and incubated with anti-UCP1 (1:10,000) and anti-PPARα (1:1000) antibodies (Abcam, Cambridge, MA, USA) at 4 °C overnight, according to the manufacturer’s instructions. Thereafter, membranes were incubated with a 1:4000 dilution of horseradish peroxidase labelled anti-rabbit (Santa Cruz, Dallas, TX, USA) conjugated secondary antibody at room temperature for 90 min. Beta (β)-tubulin (1:1000) (Cell Signalling Technology, Beverly, CA, USA) was used as a loading control to normalise protein expression. Proteins of interest were detected using a Clarity Western ECL Substrate kit (BioRad) and Chemidoc-XRS imager and quantified using Quantity One^®^ software Version 4.4 (Bio-Rad). 

### 4.5. Protein Oxidation

Protein oxidation was quantified using a protein carbonyl kit (Sigma-Aldrich) according to the manufacturer’s instructions. Briefly, samples were diluted to a protein concentration of 10 mg/mL and the carbonyl content was measured by quantifying dinitrophenyl hydrazone adducts, which form through the derivatisation of protein carbonyl groups with 2,4-dinitrophenylhydrazine. Absorbance was measured at 375 nm on a SpectraMax^®^ i3x Multi-Mode Microplate reader using the SoftMax Pro 7 Software (Molecular Devices, San Jose, CA, USA). The Bradford and RC DC protein assays are not compatible with this assay due to interference of guanidine in the samples; therefore, protein carbonyl content was normalised to protein content using the Pierce™ Bicinchoninic Acid (BCA) protein assay kit according to the manufacturer’s instructions (Thermo Fisher Scientific™, Rockford, IL, USA), and results were expressed as nmol carbonyl/mg protein.

### 4.6. Lipid Peroxidation

Lipid peroxidation in the liver was assessed using the OxiSelect™ thiobarbituric acid reactive substances (TBARS) assay kit according to the manufacturer’s instructions (Cell Biolabs, San Diego, CA, USA). Briefly, approximately 100 mg of tissue was suspended in phosphate buffered saline (PBS) containing 0.05% butylated hydroxytoluene solution (to prevent tissue oxidation) and homogenised at 25 Hz using a Tissue lyser and pre-cooled adapters (Qiagen), by alternating between 1 min in the Tissue lyser and 1 min on ice, and repeated 3 times, followed by centrifugation at 10,000 × *g* for 5 min at 4 °C to collect the supernatant. The malondialdehyde (MDA) content, a marker of lipid peroxidation, was quantified at 490 nm using a BioTek^®^ ELx800 plate reader equipped with Gen 5^®^ software (BioTek Instruments Inc., Winooski, VT, USA). The MDA content was normalised to protein content, quantified using the Bradford reagent (BioRad), and results were expressed as µmol MDA/mg protein.

### 4.7. Molecular Docking

Molecular docking studies were conducted to further elucidate the phenolic compounds that might be responsible for the increased expression of UCP1 and PPARα in CPEF-treated db/db mice. Seven of the eight major phenolic polyphenols identified in CPEF were used for the molecular docking procedure. The flavanone, eriodictyol-*O*-deoxyhexose-*O*-hexose, was excluded due to the unknown position and the identities of the sugar moieties in the molecular structure. The 2-D structures of the compounds were then drawn using ChemDraw (Version 8.0, PerkinElmer Informatics, Waltham, MA, USA) and optimised using Avogadro (Version 1.2) [62], in which partial charges were added using the Ghemical force field, followed by energy minimisation to obtain the optimal geometry for each compound.

The 3-D models of *Mus-musculus* UCP1 and PPARα were modelled using the Swiss-Model online tool [63]. Both UCP1 and PPARα were homology modelled for this study due to the lack of *Mus-musculus* crystal structures. The template used to generate the UCP1 model was PDB ID 2LCK, whilst the templates used for the PPARα model were PDB ID’s 3DZY and 1K7L. Models were validated using the MolProbity online server [64]. Using UCSF Chimera (Version 1.15) [65], the protein structures were optimised for molecular docking by removing the water molecules and all non-standard residues. The molecular docking simulation was then carried out using Autodock Vina software Version 1.2.0 [66], with the docking grid box co-ordinates presented in Table 2. Based on the Autodock Vina scoring technique, the docked complexes were ranked on binding affinity score and the root mean square deviation from the original structural pose of each compound. All complexes were superimposed to ensure the validation of the docking procedure and binding site (Appendix A). Following the molecular docking procedure, the predicted intermolecular interactions, stabilising the compound within the active site of each protein, were assessed using the ligplot application of the Ligplus software Version 2.2 [67]. The intermolecular interactions identified in each docked complex, comprising hydrophobic interactions and hydrogen bonds, were analysed and compared to CL-316,243 and fenofibrate (experimental control standards), which are compounds previously reported to experimentally activate UCP1 and PPARα expression, respectively [53,54].

### 4.8. Statistical Analysis

Data are presented as the mean ± standard deviation (SD) for 6–8 mice per group. Statistical analysis was conducted using Graph Pad Prism software (Graph Pad Software Inc. Version 7.03, San Diego, CA, USA). Statistical differences between the groups were determined by one-way analysis of variance (ANOVA) followed by Tukey multiple comparisons post-test. Non-parametric data were analysed using the Kruskal–Wallis test followed by Dunn’s multiple comparisons post-test. For all statistical tests, *p* < 0.05 were considered significant.

## 5. Conclusions

The study showed that CPEF increased UCP1 and PPARα expression in the brown adipose tissue, and increased PPARα expression in the liver of obese, diabetic female db/db mice. These results suggest that CPEF may exert its anti-obesity effects as previously reported [13] by promoting thermogenesis and fatty acid oxidation via increased UCP1 and PPARα expression. Further mechanistic studies are needed to fully elucidate the potential of CPEF to induce UCP1 and PPARα expression, and subsequently activate thermogenesis and fatty acid oxidation. Based on molecular docking and ligand interaction plot analyses, the CPEF flavanones, hesperidin and neoponcirin, showed the optimal binding affinities with UCP1 and PPARα, suggesting that the two polyphenols might be responsible for the increased UCP1 and PPARα expression exerted by CPEF. Thus, our findings support further investigations to explore the potential of developing CPEF and its major polyphenols as nutraceuticals for target-specific anti-obesity therapeutics.

## Figures and Tables

**Figure 1 ijms-24-03868-f001:**
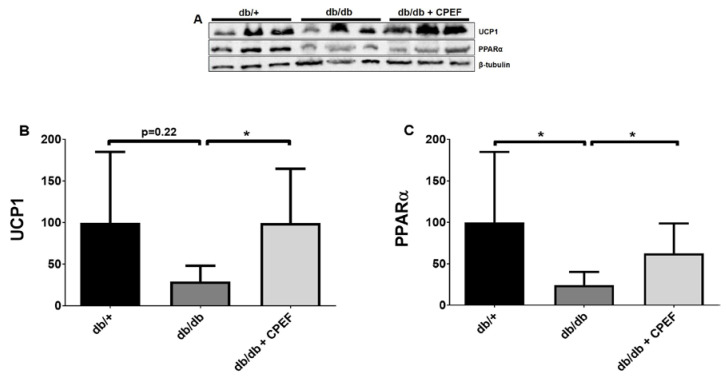
Treatment with CPEF increased UCP1 and PPARα expression in brown adipose tissue of db/db mice. (**A**) Representative western blots of UCP1, PPARα, and β-tubulin in brown adipose tissue of db/+ mice, db/db mice and db/db mice treated with CPEF. (**B**,**C**) Quantitative protein expression of UCP1 and PPARα. Mice were treated with the vehicle control or CPEF (351.5 mg/kg) daily for 28 days. Protein expression was normalised to β-tubulin and data are expressed as a percentage relative to db/+ mice, which was set at 100%. Data are represented as the mean ± SD (*n* = 6). Statistical significance between the groups is denoted as * *p* < 0.05 compared to the db/db control group.

**Figure 2 ijms-24-03868-f002:**
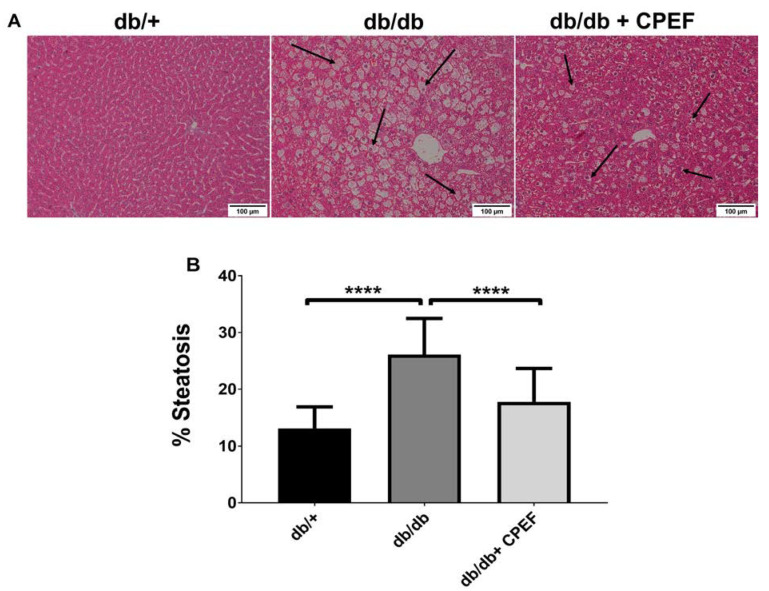
Histological assessment of the liver morphology and lipid accumulation using Hematoxylin and Eosin (H&E) staining. (**A**) Representative images of H&E-stained liver sections of db/+ mice, db/db mice, and db/db mice treated with CPEF. (**B**) Image J quantification of lipid content. Mice were treated with the vehicle control or CPEF (351.5 mg/kg) daily for 28 days. Data are represented as the mean ± SD (*n* = 6–8). Fat accumulation is indicated by black arrows. Statistical significance between the groups is denoted as **** *p* < 0.0001 compared to the db/db control group.

**Figure 3 ijms-24-03868-f003:**
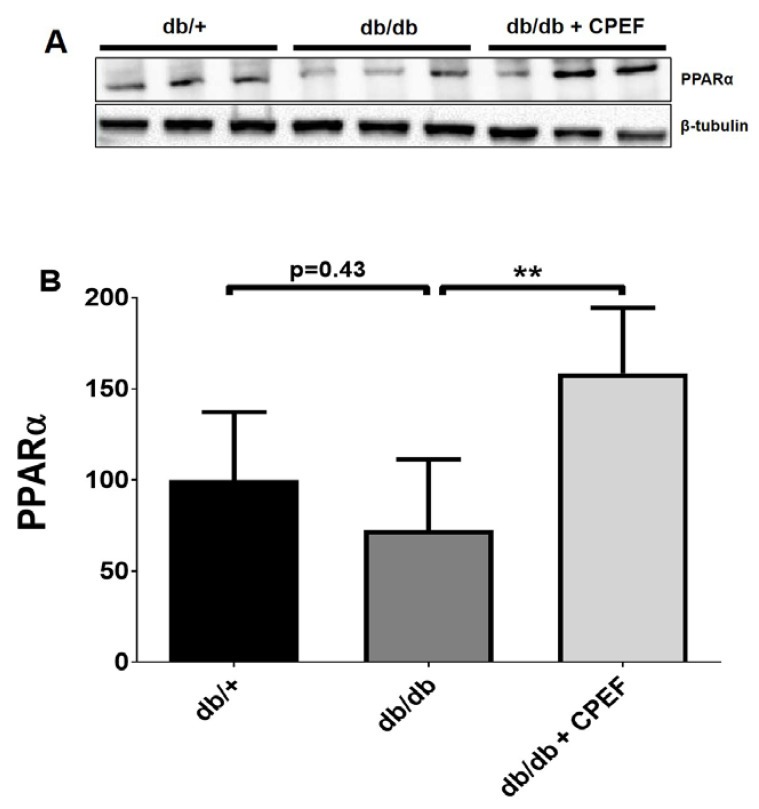
Treatment with CPEF increases hepatic PPARα expression in db/db mice. (**A**) Representative western blots of PPARα and β-tubulin in the liver of db/+ mice, db/db mice, and db/db mice treated with CPEF. (**B**) Quantitative protein expression of PPARα. Mice were treated with the vehicle control or CPEF (351.5 mg/kg) daily for 28 days. Protein expression was normalised to β-tubulin and data are expressed as a percentage relative to db/+ mice, which was set at 100%. Data are represented as the mean ± SD (*n* = 6). Statistical significance between the groups is denoted as ** *p* < 0.01 compared to the db/db control group.

**Figure 4 ijms-24-03868-f004:**
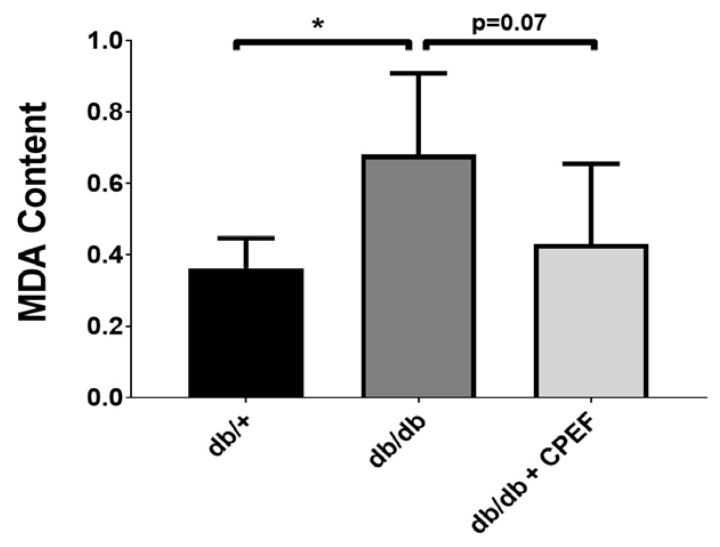
The effect of CPEF treatment on lipid peroxidation in the liver of db/db mice. Mice were treated with the vehicle control or CPEF (351.5 mg/kg) daily for 28 days. Lipid peroxidation was assessed by quantifying the malondialdehyde (MDA) content using TBARS assay in the liver homogenates and normalised to protein content (µmol/mg protein). Data are represented as the mean ± SD (*n* = 6–8). Statistical significance between the groups is denoted as * *p* < 0.05 compared to the db/db control group.

**Figure 5 ijms-24-03868-f005:**
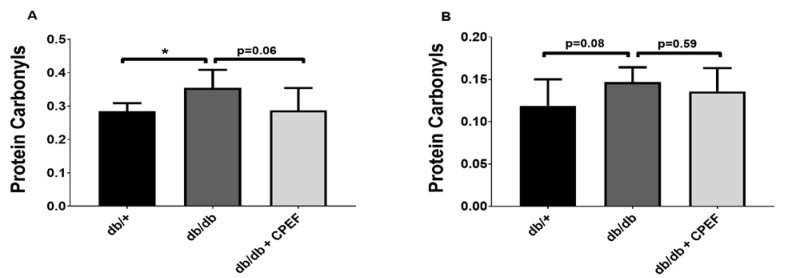
The effect of CPEF treatment on protein oxidation in the brown adipose tissue and liver of db/db mice. Mice were treated with the vehicle control or CPEF (351.5 mg/kg) daily for 28 days. Protein oxidation in the brown adipose tissue (**A**) and liver (**B**) was quantified using the protein carbonyl kit and normalised to protein content (nmol/mg protein). Data are represented as the mean ± SD (*n* = 6–8). Statistical significance between the groups is denoted as * *p* < 0.05 compared to the db/db control group.

**Figure 6 ijms-24-03868-f006:**
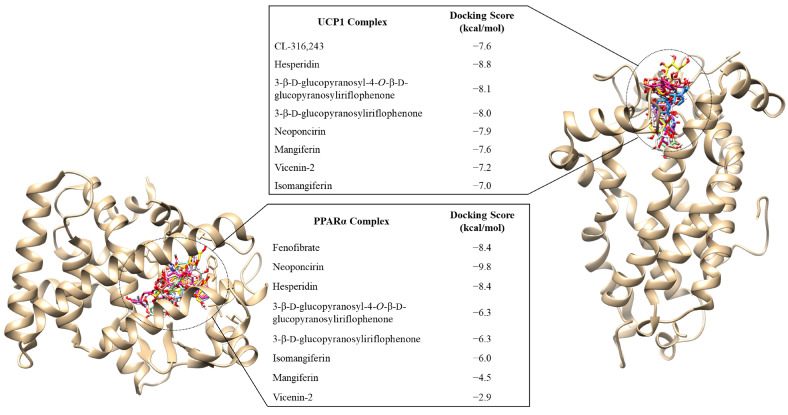
In silico binding affinity scores and superimposed docked complexes of CPEF phenolic compounds bound to target proteins, PPARα (**left**) and UCP1 (**right**). The experimental standard, fenofibrate, docked to PPARα with a binding energy of −8.4 kcal/mol, while neoponcirin and hesperidin docked with binding affinities of −9.8 kcal/mol and −8.4 kcal/mol, respectively. The known drug, CL-316,243, bound to the active site of UCP1 with a docking score of −7.6 kcal/mol, while hesperidin docked with a binding affinity of −8.8 kcal/mol.

**Table 1 ijms-24-03868-t001:** Molecular docking scores and ligand interaction plot analysis of CPEF compounds bound to UCP1 and PPARα.

Compound Name	Hydrophobic Interactions	Hydrogen Bond (Length-Å)
UCP1		
CL-316,243	Lys38, Val39, Arg40, Gln44, Gly45, Val139, Arg183, Pro179, Lys237	Glu46 (2.85), Arg140 (3.07/3.14)
Hesperidin	Gly47, Ile241, Arg140, Val39, Gly176, Asn180, Pro179, Phe240	Arg183 (2.80), Gly45 (2.78), Ala143 (2.81), Gln144 (3.22), Glu168 (2.81), Gln48 (2.88/3.10), Glu46 (2.83/3.01/2.87)
3-β-d-glucopyranosyl-4-*O*-β-d-glucopyranosyliriflophenone	Ala143, Lys38, Val39, Glu46, Arg40, Gly45, Arg140, Thr172, Glu168, Gly47, Phe240, Ile241	Thr36 (3.10), Gln48 (3.18), Gln144 (3.13/2.27)
3-β-d-glucopyranosyliriflophenone	Lys38, Val39, Glu46, Gly45, Arg40, Gly45, Ala143, Phe240	Gln48 (3.12/2.93), Thr36 (2.93), Arg140 (3.02), Asp35 (2.99)
Neoponcirin	Lys350, Val139, Pro179, Val39, Phe240, Gly47, Ile241, Leu244, Thr172, Gln247, Lys175, Val39	Arg183 (3.24), Arg140 (2.89/3.21), Arg40 (2.92), Ala143 (2.96), Glu46 (3.04)
PPARα		
Fenofibrate	Thr279, Met320, Leu321, Cys276, Ile354, Phe273, His440, Phe318, Ile317	Met355 (3.35), Ser280 (3.29), Thr283 (2.79)
Neoponcirin	Phe218, Met320, Thr283, Leu321, Ile317, Phe318, Ser280, His440, Tyr314, Ile354, Met355, Leu347, Phe351, Ile272, Phe273, Cyc276, Glu269, Thr279, Met330	Asn219 (2.80), Glu286 (3.05), Met220 (3.19/3.27)
Hesperidin	Phe218, Met320, Ile317, Leu321, Thr283, Ser280, Phe318, Tyr314, Cys276, Phe273, Glu269, Ile354, Leu347, Ile272, Leu344, Phe351, Met355, Met330, Thr279	Met220 (3.20), Asn219 (2.83), Glu286 (3.05), His440 (3.03)

**Table 2 ijms-24-03868-t002:** Gridbox coordinates of the molecular docking simulation.

Target Protein Description	Centre (X, Y, Z)	Dimensions of Grid Box (X, Y, Z)
UCP1	27.286, 34.659, 28.111	68, 72, 62
PPARα	12.619, −11.582, −29.582	24, 20, 36

## Data Availability

Data are contained within the article and Appendix A.

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
