# Peer review of "Cyclopia intermedia* (Honeybush) Induces Uncoupling Protein 1 and Peroxisome Proliferator-Activated Receptor Alpha Expression in Obese Diabetic Female db/db Mice"

_ijms, 2023, doi:10.3390/ijms24043868_

Round 1
Reviewer 1 Report
Manuscript no: ijms-2132227
Title: Cyclopia intermedia (honeybush) induces uncoupling protein 1 and peroxisome proliferator-activated receptor alpha expression in obese diabetic female db/db mice
A crude polyphenol-enriched fraction of Cyclopia intermedia was used as anti-obesity agent, and the authors found that CPEF efficiently upregulated UCP1 and PPARa expressions in the brown adipose tissue, PPARa expression in the liver and reduced heaptic steatosis. CPEF also decreased the lipid peroxidation and protein carbonylation in liver, and tended to decrease protein carbonylation in BAT. The decrease in lipid accumulation and increase in thermogenesis might be owing to the high binding affinity of compounds in CPEF with UCP1/PPARa. This is an interesting study; however, some major concerns should be clarified.
Major concerns:
Because the main hypothesis is that CPEF exhibits the role of anti-obesity, the evidence provided are not comprehensive. White adipose tissue as an important tissue for lipid metabolism is strongly recommended to be included in this study. The physiological parameters such as body weight, blood biochemistry, are recommended to include in this study as well. Morphology of BAT is recommended to be included.
Minor concerns:
Figure 2A, scale bar should be provided
Line 149, and 158, a statement for using CL 316243 and fenofibrate should be brought out
Line 316-318, Solvent for CPEF, and volume for oral administration should be provided.
Reviewer 2 Report
1. Please improve the abstract section.
2. Which PDB Id are you used for docking?
3. Figure of normal group is missing. Why not compared the normal group
to Toxic group.
4. How you said the toxic group mice liver has is i\toxicity.
5. Figure 2A not clear and All are same. Standard group are missing. Please clearly write in the figure where fat cells are deposited.
6. Which standard drug used for comparission of extract for obesity.
7. how you can said that Cyclopia intermedia is do the work on both target and show obesity activity.
8. Section 4.1, how you will quantify the written compounds is phenolics.
section 4.2, please provide the animals approval number or IAEC number.
9. To validate the molecular docking result, I suggest perform the MD simulations 50ns.
Round 2
Reviewer 1 Report
The questions have been well responsed. However, one concern still should be addressed. The reason for using female mice should be stated.
